# Coexistence of Energy Harvesting Roads and Intelligent Transportation Systems (ITS)

Domenico Vizzari [1,*], Natasha Bahrani [1] and Gaetano Fulco [2]

1 COSYS—SII Laboratory, University Gustave Eiffel, Campus of Nantes, 4340 Bouguenais, France
2 DIIES—Department of Information Engineering, Infrastructure and Sustainable Energy, Mediterranea University of Reggio Calabria, 89124 Reggio Calabria, Italy
* Correspondence: domenico.vizzari@univ-eiffel.fr

**Abstract:** Intelligent systems, the Internet of Things, smart factory, and artificial intelligence are just some of the pillars for the 4th industrial revolution. Engineering is the driving force behind this new industrial renaissance and transportation plays a leading role for the new challenges in mobility needs. In this scenario, intelligent transportation systems (ITS) represent an innovative solution for various transport issues, such as traffic congestion, air pollution, long travel time, and accidents. In parallel, transportation is going through a novel way of thinking for road pavements: a multi-functional infrastructure able to harvest energy and exploiting the solar radiation or the traffic load. As the main hurdle in ITS is to find reliable energy sources, the energy harvesting roads could be a great step in installing and managing ITS as an electricity supplier. The aim of this paper is to review the key elements of ITS and energy harvesting pavements, and investigate their coexistence. This paper describes different harvesting techniques that could be used to power various ITS solutions. A case study evaluates the power output of a road section equipped with a solar road, piezoelectric material, and thermoelectric generators. Finally, the coexistence between ITS and energy harvesting pavements is critically evaluated, taking into account the advantages and disadvantages.

**Keywords:** energy harvesting; road infrastructure; intelligent transportation systems

## 1. Introduction

The intelligent transportation systems (ITS) deal with advanced applications to provide innovative services for the different modes of transport. Various technologies of ITS are defined for the different modes and infrastructures of transport to improve their overall functionality. Generally, ITS are used to improve traffic and mobility management, and to provide added services to the transportation system. ITS fit very well in the 4th industrial revolution, which comprises the introduction of cyber-physical systems and the integration of the sensor data collection and processing. The concepts of integration data analysis, data visualization, decision-making, and predictive analysis aid in the intelligent services, which are essential for the enterprises [1].

Tying down the concept of the 4th industrial revolution and the various ITS applications could highly contribute to the efficiency of road management and safety.

The widely-used applications of ITS include traffic management and control, tolling, road pricing, road safety, and law enforcement, public transport travel information and ticketing, driver information and guidance, freight and fleet management, and vehicle safety. With the increased usage of ITS services, the power supply becomes crucial, especially in locations where it is difficult to provide electricity, or where the existing surfaces could be exploited for other purposes. In this scenario, the energy harvesting pavements could be an efficient way to meet the power needs of ITS in remote areas, decentralizing the expansion of energy sources and avoiding long power lines with consequent energy losses.

The energy harvesting pavements consist of technologies able to exploit external sources to produce electric power or to stock heat energy through the road infrastructure.

The sources with the highest potential are the solar radiation and the vehicle load. The solar radiation can be directly converted into electricity by the solar cells and thermoelectric generators, or stocked in the form of thermal heat through a heat-transfer fluid, according to the principle of the solar thermal technology. The vehicle load can be converted into electricity by piezoelectric materials integrated into the pavement, or by speed bumps via electric generators.

It is worth noting that the sources mentioned above are intermittent and strictly dependent on the location of the road section, the climate condition, and the traffic flow. Therefore, the question is: can ITS and energy harvesting pavements work in harmony? How to overcome some intrinsic problems of coexistence, due to the limitations for the implementation of some energy harvesting pavements? How to ensure the energy supply without compromising the ITS operation conditions? What are the positives, negatives, and constraints of this coexistence? In this paper, we first discuss the key elements of ITS technologies and their various applications, following with the energy consumption of the ITS technologies. We then present the key elements of the energy harvesting technologies and the working principle for the energy storage. Moving from real traffic and weather data, we have written a case study of a road section equipped with energy harvesting systems. The objective is to evaluate the power output of the aforementioned road and to understand if the electricity can answer the power needs of the ITS. Finally, the coexistence between ITS and energy harvesting pavements is critically evaluated, in terms of the advantages and disadvantages.

## 2. Key Elements and Applications of the Intelligent Transport Infrastructures

The idea of intelligent transport infrastructures is to overcome the existing problems in transportation (traffic movement and congestion, public transport crowding, off-peak inadequacy of public transport, difficulties for pedestrians, parking difficulties, environmental impact and traffic noise) thanks to the next generation ITS technologies. With ITS, it is possible to interoperate the different fields of transport, such as traffic management, operations, control, and policies. ITS are able to combine the road systems with the vehicles and drivers, to support and solve some transport problems. There are multiple generations of the ITS, depending on their level of development [2]:

- ITS1.0 (2000): First Generation—One-way infrastructure technology;
- ITS2.0 (2000–2003): Second Generation -Two-way communication technology;
- ITS3.0 (2004–2005): Third Generation- Automated vehicle operations and automated, interactive system operations and system management;
- ITS4.0 (2006–2011): Fourth Generation- Multi-modal incorporating personal mobile devices, vehicles, infrastructure, and information networks for system operations, as well as personal contextual mobility solutions.

Mainly, ITS consist of on board units (OBUs), road side units (RSUs), a central system, and personal systems. From the third generation onwards, ITS have also evolved into cooperative intelligent transport (C-ITS), which enables the service provision and use of dynamic data/information from other entities with the same function. This could be vehicle to vehicle, vehicle to infrastructure, or vice versa. C-ITS allows two or more ITS sub systems to cooperate and communicate securely (Figure 1). ITS also consist of telematics systems, which are mostly used for tolls concerning trucks and charging fees [3].

In general, ITS are vast fields, mainly concerning the safety and efficiency of the transport infrastructure. In the following sub-sections, some of the main applications of ITS are described.

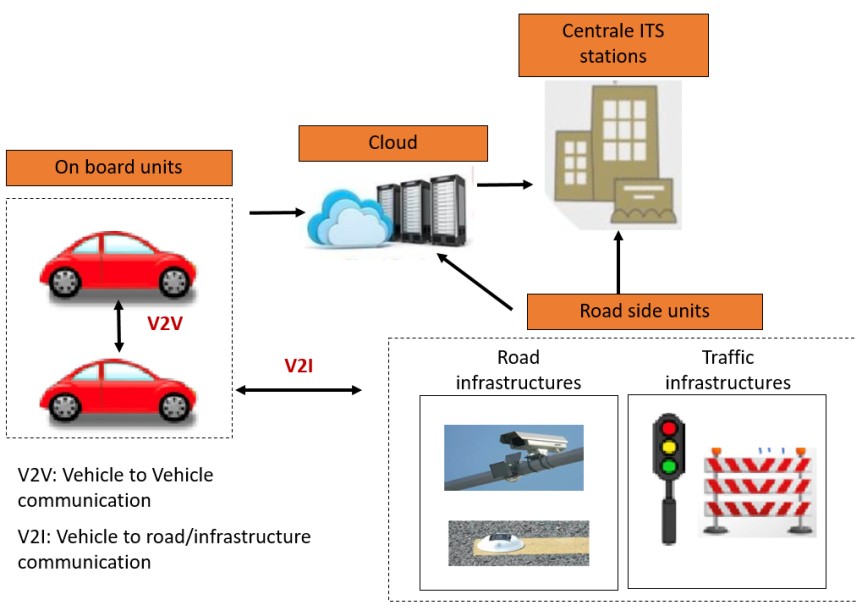

**Figure 1.** Architecture for ITS.

### 2.1. Advanced Traffic Management Systems

Traffic management systems are essential in safety and mobility management. They collect the real-time data of traffic through different cameras and sensors; then the data are sent to the transportation management data centre. Some of the devices designated to this application include queue detectors and CCTV (closed-circuit television) cameras for surveillance. Specific types of road side units (RSUs) are also used for the traffic monitoring on the road network. The RSUs collect the data and relay the information to the data centre of the traffic manager. These systems detect and map the traffic (cars, motorcycles, heavy good vehicles, and buses) and observe the behaviours, such as peak hours, speeds distance between vehicles, etc.

### 2.2. Electronic Payment Systems

Another important application of ITS is the automatic toll collection, which does not require any stopping of the vehicles, in order to collect the toll. It allows the smooth flow of traffic and reduces the traffic congestions. It is an evolution of technology, based on the vehicle positioning system techniques for a debit transaction VPS system on the freeway [4]. Low power radio frequency identification (RFID) technology is also used to collect the information of the vehicle and to detect the amount to debit. The RFID uses tags and readers to collect the information [5].

### 2.3. Advanced Vehicle Information and Navigation System

The advanced ITS technologies provide GPS-based solutions combined with the general packet radio service (GPRS) and global system for mobile communication (GSM) systems to find the location and send the information to the server. These could be used to monitor and track the vehicle in real time and determine the travel time for the destinations [6]. The data could be exploited to relay important information to the travelers through dynamic message signs (DMSs) or through portable changeable message signs.

### 2.4. Advanced Commercial Vehicle Operations

These operations concern the administration and regulation control of the transport infrastructures, which involve the use of the information and communication technology (ICT) to perform checks and documentation on the commercial services. ICT allows vehicles, such as trucks and buses to have their credentials and status checked with minimum time or no interruption. Some examples of this technology are the automated border crossing and

automated commercial vehicle administration documents [7]. In this regard, the heavy vehicle loading control could also be useful to monitor the maximal allowable traffic on the highways. The weigh-in motion (WIM) stations are installed in various locations to control and regulate the allowable loads. These WIM stations are equipped with sensors, electromagnetic loops and piezoelectric quartz to gather the information concerning the weights.

*2.5. Disaster Management System*

The disaster management system is essential for the safety of the people and the health of the transport infrastructure. It is important to install a system that could benefit from the infrastructure when there is a disaster. This system includes the collision notification and avoidance system, which is able to detect and report the incident. Other components are the emergency vehicle warning, intersection collision warning, motorcycle approaching indication, and the optimal traffic light speed advisory [8].

Merenda et al. [9] propose an innovative platform, based on a network of wireless, low power and renewable-energy-fed sensor units. The platform is able to provide important information on the structural health status of the infrastructure and it gives useful data for maintenance. Furthermore, it works as a decision support tool for emergency management and post-disaster assessments.

*2.6. ITS for Road Health Monitoring*

For an efficient transport infrastructure, it is essential to regularly monitor and assess the condition of the infrastructure. There are various structural health monitoring methods and studies in regard to the monitoring the roads. For instance, there is a big boost in using indirect structural health monitoring (SHM) or techniques, such as ITS for monitoring the road health. Various technologies, such as wireless sensor networks and embeddable sensors are being used to monitor the properties of the infrastructure [10–13].

Another important project that depicts the use of ITS and road monitoring is Safe Strip, which is a European project that focuses on the ITS solutions for self-explaining and forgiving road infrastructures [14]. The project uses an economical sensory system, which could be integrated in road pavements, and it provides useful information to the road users. Different enabling technologies could be used to integrate Safe Strip, such as communication and sensors technologies, inductive loops, video surveillance, data collection, and management technologies.

*2.7. Energy Consumption of the ITS Technologies*

Energy consumption of the ITS technologies is one of the main issues when it comes to the installation of ITS. Depending on the type of applications and the type of technologies, the power consumption varies drastically.

The two main technologies used by ITS are the sensing and the wireless sensor network technologies (WSN). Sensor technology is a broader term that usually is used for sensing and processing a parameter; this is often packaged with other technologies to deliver different applications. Moreover, WSN are usually low memory and low processing power technologies with a limited communication capacity. They require a base station to obtain the information, to process it, and relay it further. WSN often involve sophisticated techniques and protocols that allow the seamless data transfer with the minimum interruption of the data communication. WSN could be used to cover a large-scale network [15]. The main difference between the regular sensors and WSN is the use of wireless sensors for the transmission and the reception of the information. Due to their nature of not using wires [16], WSN require batteries and efficient energy sources. Whereas the regular sensor technologies (i.e., strain gauges and wired accelerometers used for pavement and traffic monitoring), depend on the wired power supply and they could be installed where there is a power source.

It is vital to have an adequate support for these devices, which includes the arrangements for the power sources where it is necessary. The existing methods for power sources are given below:

- Main grid power supply: Electric power distribution and communication is the preferred electric power supply for the ITS services and applications;
- Battery powered devices: Recent ITS innovations and applications have been introduced with the small self-powered systems. These technologies are especially useful for remote zones where direct supply is not possible;
- Solar: For locations that are well exposed to the solar radiation, where direct electric supply is not possible.

As there are vast types of applications and types of energy consumption concerning ITS, some reference values are presented in Table 1.

**Table 1.** Energy consumption of the ITS technologies.

| ITS Technologies | Power Consumption [W] |
| --- | --- |
| Sensor technology | 0.5–25 W [17] |
| CCTV cameras | 24 W [18] |
| WIM technology | 2.97–9.57 mW [19] |
| Wireless sensor technology | 0.7–0.6 mW [20,21] |
| RFID systems | Up to 500 mW [22,23] |
| RSUs/Data acquisition systems | 300 mW [24] |
| DMS/Portable chargeable message signs | 300 W/m$^2$ [25] |

## 3. Key Elements of the Energy Harvesting Pavements

The main requirement from the energy harvesting pavements for the ITS is the energy supply. The systems that are able to answer the electric demand, are as follows (Figure 2):

- solar roads [26];
- piezoelectric pavements [27];
- TEGs (thermoelectric generators) [26];
- speed bumps called TEDHs (traffic energy harvesting devices) [28].

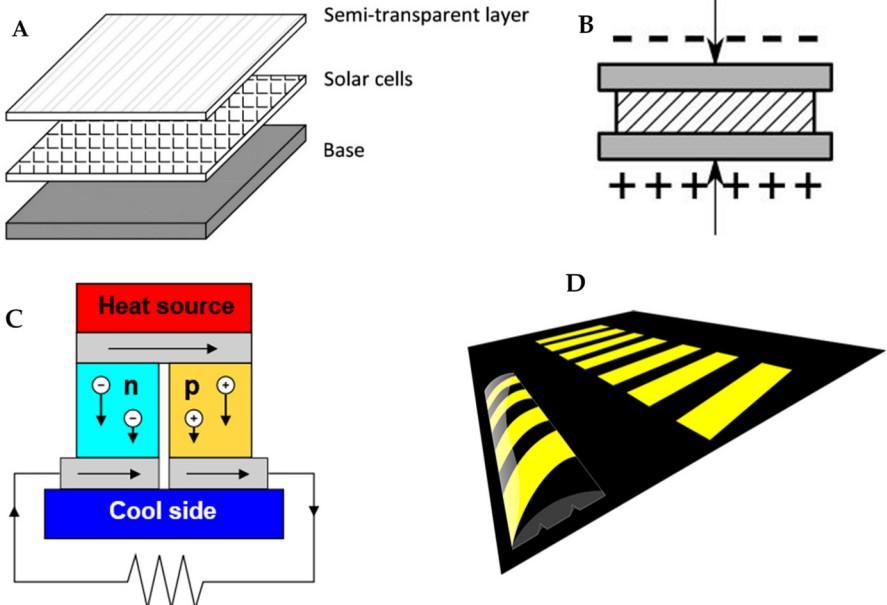

**Figure 2.** Examples of energy harvesting roads. (**A**) solar road; (**B**) piezoelectric material; (**C**) thermoelectric generator; (**D**) speed bump (traffic energy harvesting device).

The solar roads consist of solar cells directly imbedded in a semi-transparent layer. Commercial solar cells have an efficiency in the range of 15–22%. Due to the phenomenon of the reflection/absorption of the sunlight from the semi-transparent layer, as well as the presence of dust and a non-optimal tile angle, the efficiency undergoes a further reduction from 10 up to 80% [29]. The remaining energy is lost, in the form of heat. This latter causes a change in temperature of the road infrastructure, leading to a conduction process from the surface to the interior of the road [30]. The performance degradation of the solar road is related to the aging of the semi-transparent layer and of the solar cells. Over the years, the material could turn yellow because of the exposition to the UV radiation, while the solar cells experience a degradation rate of around 1%, per year [31]. Both phenomena heavily contribute to the efficiency loss of the solar road.

The thermoelectric generators (TEGs) are able to convert the thermal energy into electricity, exploiting the difference in temperature between the road and the underground. TEGs are based on the Seebeck effect, a physical phenomenon where the difference of temperature between the junctions of a thermocouple is converted into electricity. If a voltage is applied to the junctions, the result is a difference in the temperature between them (Peltier effect). Finally, if a current is flowing through an electrical conductor with a temperature gradient, the heat is released or absorbed, based on the nature of the material and the direction of the current flow (Thomson effect) [32]. TEGs are distinguished in (i) liquid heat dissipation TEGs, using heat sinks to maintain the low temperature at the cold side of the TEG; (ii) solid heat dissipation TEGs, in which the liquid heat dissipation is replaced by a copper plate, and (iii) thermoelectric cement-based composites, in which the Seebeck effect is given by the charge carriers moving from a hot point to a cold point within the cement materials containing carbon fibers [33,34].

In terms of the performance deterioration, it can significantly increase when the TEG is subjected to thermal cycling [35]. Cerqueira Veras et al. [36] performed thermal cycling from 20 to 40 °C every 15 min. Following 548 cycles, they observed that the internal resistance was increased by 9.8%, while the Seebeck coefficient, the thermal, and the electric conductivity were reduced by 3.9%, 8.6%, and 9.6 %, respectively. In the same range of temperatures, Tenorio et al. [37] alternated the hot and cold sides across a TEG. Following 127 cycles, the open circuit voltage was reduced by 9.99%, the Seebeck coefficient by 9.57%, while the internal resistance was increased by 11.07%. In the literature, most of the other tests are performed at a higher range of temperatures (over 200 °C), which are not replicable for road applications.

The piezoelectric roads can convert the vehicle load into electricity, thanks to piezo-electric crystals imbedded into the asphalt [38]. During the manufacturing process, the polar domains of the piezoelectric element are randomly oriented. At this stage, if an electric field is applied just below the Curie point, the domains will tend to align with the electric field. When the wheel of the vehicle goes through the section equipped with the piezoelectric material, this latter undergoes a displacement inside electric poles. The result is an electric potential of the two poles [39]. The piezoelectric degradation is due to depolarization or cracks: depolarization is defined as a shift of the polar domains from the initial orientation; the cracks can appear after fatigue cycles and they can develop through the thickness, along the length, or the interstitial surfaces of the piezoelectric material. Usually, depolarization is correlated to cracking and both contribute to the efficiency loss of the piezoelectric materials [40].

The TEDHs are able to convert the weight force of the vehicles into electric power via electric generators [41]. Depending on the mechanical element for the energy conversion, there TEDHs can be classified in: (i) hydraulic TEDHs, in which a hydraulic turbine converts the kinetic energy of the fluid into a mechanical work; (ii) pneumatic TEHDs, characterised by gas or air, instead of incompressible fluid; (iii) mechanical TEDHs, where the vertical displacement of the speed bump is directly converted into the rotational movement inside the generator through rods, crankshafts and gears; (iv) electromagnetic TEDHs, in which the kinetic energy of the vehicle is converted into a relative movement between magnets

and coil, in order to generate electricity, according to the Faraday law [42]. The power generated by each type of technology is given in Table 2.

**Table 2.** Power generated by the different energy harvesting techniques.

| Solar Road | Thermoelectric Generators | Piezoelectric Road | TEDHs |
|---|---|---|---|
| 48 [43]–106 [44] W/m$^2$ | 0.09 [45]–0.41 [43] W | 0.0015 [39]–0.14 [43] W/m$^2$ for vehicle | 20 [42]–200 [46] W/vehicle |

The research shows a growing interest in harvesting technologies. Sodano et al. [47] formulated a model of a power harvesting system given by a cantilever beam with piezo-electric patches. Anton and Sodano [48], implemented piezoelectric material to recharge a watch battery in less than 1 h. Sodano et al. [49] investigated the ability of different piezo-electric devices to recharge various capacity nickel metal hydride batteries, under different excitation conditions. This work provides a comparison of the different technologies with tables for the efficiency and charging time.

Anton et al. [50] present a multifunctional piezoelectric energy harvesting system. It is composed of piezoelectric layers for the power generation, thin-film batteries for the energy storage, and a central metallic substrate layer. The charge/discharge results demonstrate the ability of the self-charging structure to simultaneously generate and store electrical energy. In terms of the current, the system generates a current of 0.08 mA, corresponding to an average power of around 0.306 mW, during charging.

Anton et al. [51] exploited solar, piezoelectric, and thermal energy through a single device. The system is able to charge a 1 mAh battery in 20 min using solar energy, in 40 min using thermal energy, and in 8 h using vibrational energy.

Regarding the applications on highways, in the literature, there are experimental applications of piezoelectric materials installed in a roadway speed bump. For example, Song et al. [52] designed a speed bump with 40 piezo-generators installed directly in a module with dimensions of $30 \times 20 \times 8$ cm$^3$. The speed bump generates a maximum power of 4.08 W, considering a medium-sized passenger car running at a speed of 30 km/h.

Jung et al. [53] developed an energy harvesting module for roadway applications composed of flexible PVDF (polyvinylidene fluoride) polymers. The system consists of 60 piezoelectric elements connected in parallel, with a better performance in terms of the total power and overall impedance. The module generates an instantaneous power output up to 200 mW across a 40 kΩ resistor, at the speed of 8 km/h and a weight of 250 kgf.

## 4. Working Principle for Energy Storage Systems

Due do the discontinuity of the renewable sources, the harvested energy needs to be stocked through batteries, in order to supply the electricity to the ITS.

The growing diffusion of systems with distributed sensor networks, often in areas difficult to reach with the power supply of the electricity grid, has led the world of research to develop systems for the production of electricity and storage systems, to make the sensor network autonomous, with regard to its power supply.

Among the many techniques to produce electricity in isolated areas, they range from consolidated ones, such as photovoltaics and wind, to the most innovative production techniques that exploit the piezoelectric effect and the temperature gradient of the materials. At present, the challenge is to store this energy and deal with the intermittent renewable energy sources. The objective is to ensure the continuity of the energy supply [54].

The excess of electric energy produced by solar cells, TEGs, speed bumps, and piezo-electric materials, can be stored in backup batteries and used to power the ITS when required (Figure 3).

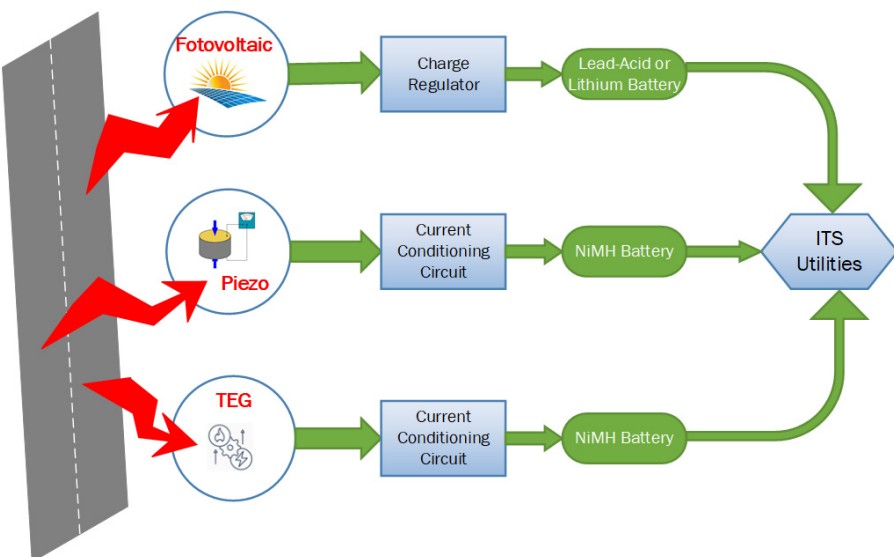

**Figure 3.** Schematic representation of the energy storage systems for ITS utilities, through energy harvesting pavements.

For the solar cells, the most common energy storage systems are lead-acid batteries or lithium batteries, characterized by a higher performance, but with less robustness [55]. It is important that the specific charge regulators are well-designed, according to the values of the currents and the voltages [56].

In the case of piezoelectric materials, the stocking of energy is more challenging, because the vibration induced by the traffic load is not constant. As a result, the alternating current peaks are generated. To convert this energy into a direct current that can be stored in accumulators, an appropriate current conditioning circuit is essential.

Anton et al. [51] proposed a two-stage power conditioning circuit with a buck-boost converter as the first stage and a linear regulator as the second stage for the TEGs. The buck-boost converter is necessary for matching the impedance of the maximum power, while the linear regulator controls the output voltage for charging the battery. The circuit for the piezoelectric energy harvesting, includes a rectifier and a linear regulator. The rectifier converts the alternate current in a continuous current, while the regulator interfaces directly with the battery.

Small and high-performance nickel metal hydride batteries can be used with piezoelectric materials and TEGs because they have a high charge density and they do not require any type of charge controller or voltage regulator. In general, this type of battery requires a simple circuit with a rectifier and a capacitor [47]. Ottman et al. [57] proposed the use of an adaptive step-down DC-DC converter to maximize the power output from the piezoelectric generator.

## 5. Case Study

The case study refers to 10 m of a hypothetical highway located in Nantes (France). The road is a flexible pavement with two travel lanes in each traffic direction and a width of 3.75 m.

The objective of the case study is:

1. To evaluate the harvested energy of a highway section equipped with a solar road, piezoelectric material, and a thermoelectric generator;
2. To propose a technical solution for stocking the electric power, using batteries and/or super capacitors;
3. To better understand if ITS can coexist with the energy harvesting pavements.

From right to left, the road section is equipped with thermoelectric generators on the shoulder, with piezoelectric material in the right lane, and a solar road in the fast lane. The logic behind this choice is that the shoulder is not subjected to the traffic load, except in exceptional cases, the right lane is more stressed by the traffic vehicle, especially

truck loads, while the fast lane has a lower traffic density and the average vehicle speed is higher. Consequently, the thermoelectric generator is not exposed to the traffic hazard, the piezoelectric material is more excited by the heavy traffic, and the solar road is less overshadowed (Figure 4).

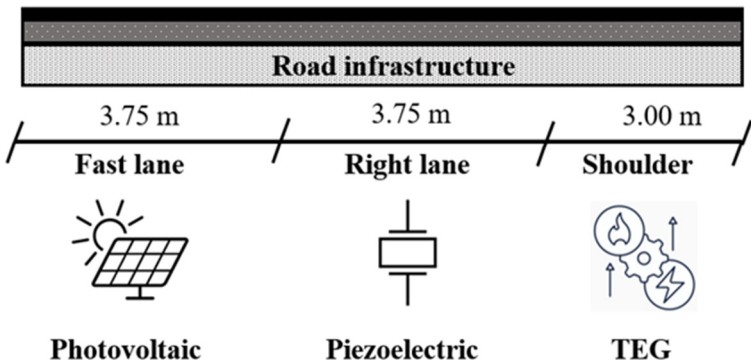

**Figure 4.** Sketch of the road section equipped with energy harvesting technologies.

In this case study, we did not consider the TEDHs because the hypothetical road section is a highway, hence the speed bumps could not be installed.

### 5.1. Electrical Output of the Solar Lane

The solar lane consists of solar cells embedded into a multilayer matrix of resin and polymer. The coating is given by transparent silicone resin and glass particles with small particles sizes. This solution, patented by Colas Company [58], can be directly installed on existing pavements. The electrical output is calculated through the model of Colagrande and D'Ovidio [59]:

$$E = [(\eta \cdot A \cdot G)(1 - \varepsilon)] \tag{1}$$

$$\varepsilon = \frac{D(\varphi) \cdot \bar{l}}{1000} \tag{2}$$

$$D(\varphi) = \frac{\varphi}{v} \tag{3}$$

$$\varphi_{max} = \sqrt{\frac{a'_m \cdot n}{2 \cdot K \cdot S}} \cdot 3600 \tag{4}$$

where: $\eta$ is the efficiency of the solar pavement; $A$ is the surface of the solar pavement (m$^2$); $G$ is the global horizontal irradiance (kWh/m$^2$/day); $\varepsilon$ is a parameter which takes into account the overshadowing caused by the passage of the vehicles; $D(\varphi)$ is the vehicular traffic density (vehicle/km); $l$ is the average length of the vehicles; $\varphi$ is the vehicular flow (veh/hour); $v$ is the average vehicular speed (km/h); $\varphi_{max}$ is the maximum vehicle flow (veh/hour); $a'_m$ is the average deceleration (m/s$^2$); $n$ is the number of vehicles per convoy; $k$ is a safety coefficient; $S$ is the average spatial spacing between cars (m). Assuming: *GHI* = 3.377 kWh/m$^2$/day [60]; $\eta$ = 0.14 [61]; $a'_m$ = 3 m/s$^2$; $n$ = 1; $k$ = 1.2; $S$ = 6 m; than $\varphi_{max}$ is 1643 veh/hour. Considering $v$ = 100 km/h and $l$ = 5 m, than $D(\varphi_{max}) \approx 16$ veh/km and $\varepsilon$ = 0.082. On average, the electrical output along 35 m$^2$ of the solar road is 16.1 kWh/day.

### 5.2. Electrical Output of the Piezoelectric Lane

The piezoelectric lane consists of lead-zirconate-titanate (PZT), widely used for commercial applications. The electrical output of the piezoelectric lane is calculated from the technology proposed by Xiong and Wang [39]. They installed six piezoelectric energy harvesters (PEHs), placed along the wheel paths of the vehicle, in order to maximize the hitting rate. Each PEH is protected with an insulating plastic, able to resist the impact of the traffic load and to uniformly transfer the applied stress of the vehicles to the PEH. The average power output is 0.0082 mW/kN, while the voltage is between 298 and 486 V, depending

on the axle loading. For this case study, we propose to convert the traffic characteristics in ESALs (equivalent single-axle loads), which represent the number of repetitions of an 18,000 lb single-axle load applied to the pavement on two sets of dual tires.

In terms of ESALS, the average power output is the accumulated ESAL for each category of axle-load, as given by the formula:

$$ESAL_i = f_d G_{jt} AADT_i 365 N_i F_{Ei} \tag{5}$$

where: $f_d$ is the deign lane factor; $G_{jt}$ is the growth factor for a given growth rate *j* and design life *t*; *AADT* is the first-year annual average daily traffic for the axle category *i*; $N_i$ is the number of axles on each vehicle in category *i*; $F_{Ei}$ is the load equivalency factor for the axle category *i*.

The traffic characteristics refer to the real data from road sensors installed in the right line of an operational highway in Nantes [62], therefore $G_{jt}$ and $f_d$ are assumed to be equal to 1. Under the hypothesis that the road is a flexible pavement, subjected to single axle loads, with a final serviceability index of 2.5 and a pavement structural number SN of 5, the load equivalency factors can be extracted from Tables D.4, D.5 and D.6, and Appendix D of the AASHTO design procedure [63]. The calculated ESAL for the vehicle and the maximum weight axles is given in Table 3.

**Table 3.** Calculated ESALs for the different vehicles and axles.

| Type of Vehicle | Max Weight Axle | AADTi | % for Each Type of Vehicle | $F_{E\_i}$ | Ni | $ESAL_{Si}$ (1st Year) |
|---|---|---|---|---|---|---|
| Cars | Na | 321 | 65% | Na | Na | Na |
| 2 axle trucks | 20,000 + 14,000 | 4 | 0.8% | 1.87 | 1 | 2730 |
| 3 axle trucks | 20,000 + (14,000+14,000) | 1 | 0.3% | 2.6 | 1 | 949 |
| 4 axle trucks | 20,000 + 20,000 + (17,000 + 17,000) | 8 | 1.6% | 4.11 | 1 | 12,001 |
| 5 axle trucks | 20,000 + (17,000 + 17,000) + (17,000 + 17,000) | 11 | 2.3% | 3.69 | 1 | 14,815 |
| T2S3 | 20,000 + (17,000 + 17,000) + (14,000+14,000+ 14,000) | 145 | 30% | 3.197 | 1 | 169,201 |

The total of ESALs for the first year is 199,697, corresponding to a cumulative power output of 131 W/year (or 0.36 W/day). Assuming to install 10 piezoelectric harvesters in a series along the right lane (one piezoelectric harvester per meter), the total electrical output is 3.6 W/day.

*5.3. Electrical Output of the TEG Lane*

For the TEG, we chose the technology proposed by Datta et al. [45]. The thermoelectric material is bismuth telluride, characterized by a low thermal conductivity, to avoid most of the heat flows from the hot to the cold side of the TEG. The TEG module is a circuit containing thermoelectric materials, which is able to generate electricity, according to the Seebeck effect. The module is composed of two types of semiconductors, called the n-type (negatively charged) and the p-type (positively charged).

The TEG is installed below the pavement surface. The heat of the road is transferred to the hot side of the TEG through a conductive element in copper, while the cold side of the TEG is maintained at a lower temperature, thanks to a heat sink placed in the soil.

The generated voltage of the TEG is given by the formula:

$$V = N\left(0.0002 \times 1.004^{\Delta T}\right)\Delta T \tag{6}$$

where:

- $N$ is the number of the thermoelectric elements in the TEG;
- $\Delta T$ is the difference in the temperature between the hot and the cold sides of the TEG.

The voltage of the TEG is maximized using an external resistance RL, according to the formula:

$$V_L = \frac{V \times R_L}{(R_L + R_{int})} \tag{7}$$

The internal resistance $R_{int}$ of the TEG is back-calculated from the measured electric power using the equations:

$$I_L = \frac{V_L}{R_L} \tag{8}$$

$$P_L = V_L I_L \tag{9}$$

where $P_L$ and $I_L$ are the power and the current output of the TEG, respectively.

Taking into account the field experiment results of Datta, the $R_{int}$ is 23.4 Ohm.

For this case study, the temperature of the top layer pavement refers to the monthly average measurements of an existing highway [64], while the temperature of the soil is calculated through TRNSYS software, using the "Type 77". The latter is based on the formula of Kasuda [65], for which the temperature of the ground is a function of the time of year and the depth below the surface. Assuming the soil is composed of sand, the input data for Nantes are summarised in Table 4:

**Table 4.** Input data for the case study.

| | |
|---|---|
| Annual average ambient air temperature | 11.7 °C |
| Amplitude of the surface temperature (maximum air temperature minus minimum air temperature) | 8.61 °C |
| Time difference between the occurrence of the minimum surface temperature and the beginning of the calendar year | 12 days |
| Depth below the surface | 0.17 m |
| Soil thermal conductivity | 2.592 kJ/h mK [66] |
| Soil density | 1200 Kg/m$^3$ [66] |
| Soil specific heat | 1.25 kJ/ Kg K [67] |

Due to the heat loss phenomenon between the copper plate and the asphalt, the difference in temperature (ΔT) between the two sides of the TEG is, on average, 31% of the ΔT between the asphalt and soil. Table 5 shows the voltage and power output calculated for each month, corresponding to the respective temperature data.

**Table 5.** Calculated voltage and power corresponding to each month.

| Month | Soil Temp [°C] | Pavement Temp [°C] | ΔT Pavement-Soil [°C] | ΔT TEG [°C] | V [Volt] | V$_L$ [Volt] | I$_L$ [Amp] | P$_L$ [mW] |
|---|---|---|---|---|---|---|---|---|
| January | 3.8 | Na | Na | Na | na | Na | Na | Na |
| February | 4.7 | Na | Na | Na | Na | Na | Na | Na |
| March | 7.4 | 15.6 | 11.8 | 3.6 | 0.424 | 0.313 | 0.013 | 8.4 |
| April | 11.3 | 23.3 | 18.6 | 5.7 | 0.674 | 0.497 | 0.021 | 21.2 |
| May | 15.3 | 25.2 | 17.8 | 5.5 | 0.644 | 0.475 | 0.020 | 19.3 |
| June | 18.3 | 30.9 | 12.6 | 3.9 | 0.452 | 0.334 | 0.014 | 9.5 |
| July | 19.6 | 33.3 | 13.7 | 4.2 | 0.495 | 0.366 | 0.016 | 11.4 |
| August | 18.6 | 30.4 | 11.7 | 3.6 | 0.421 | 0.311 | 0.013 | 8.3 |
| September | 15.9 | 24.8 | 8.9 | 2.8 | 0.320 | 0.236 | 0.010 | 4.8 |
| October | 11.9 | 18.8 | 6.8 | 2.1 | 0.245 | 0.181 | 0.008 | 2.8 |
| November | 7.8 | 15.3 | 7.6 | 2.3 | 0.270 | 0.199 | 0.009 | 3.4 |
| December | 5.0 | 10.2 | 5.2 | 1.6 | 0.185 | 0.136 | 0.006 | 1.6 |

On average, the power output is 9.1 mW, corresponding to 218.4 mWh/day (9.1 mW per hour and multiplied by the 24 h and two generators per meter, and the road is around 10 m). Assuming to install 20 TEGs in a series along the shoulder (two TEGs per meter), the total electrical output is 2.18 Wh/day.

### 5.4. Sensitivity to the Traffic

The traffic affects the power output of the energy harvesting pavements. In general, the increase of traffic reduces the power output of the solar pavements because of the vehicles' shadow; inversely, the power output of the piezoelectric pavements increases. For the TEGs, there is a lack of studies tackling the influence of the traffic. However, some authors investigated the influence of the traffic on the road surface temperature. They observed a temperature variation of up to 1.5 °C because of the heat fluxes of the vehicles. The TEGs could benefit from this phenomenon. For instance, increasing the data of the ΔT pavement-soil by 1.5 °C in Table 5, the electrical output could increase by up to 1.67 times.

To quantitatively assess the sensitivity to the traffic for the solar and the piezoelectric lanes, we calculate the power output at different traffic flows [68], using the same approaches of Sections 5.1 and 5.2. For the piezoelectric lane, we assume that: (i) the heavy vehicles represent 35% of the total; (ii) the type of vehicles are grouped in percentage, as shown in the fourth column of Table 3; (iii) the traffic flow refers to two peak hour periods per day (from 7 am to 9 am and 5 pm to 7 pm); (iv) the traffic flow outside of these hours, are not taken into account for the calculation of the electrical output of the piezoelectric lane.

When the traffic flow increases from 121 to 2930 veh/hour, the electrical output of the solar lane drops by 22%; on the contrary, the electrical output of the piezoelectric lane increases 24 times (Table 6). This result is coherent with the operating principle of the piezoelectric materials, which generate a power output for each passage of the vehicles' axles.

**Table 6.** Electrical output of the solar and piezoelectric lanes at different traffic flows.

| Flow (Veh/Hour) | Mean Speed (Km/h) | Density (Vehicles/Km) | Electrical Output Solar Road (kWh/Day) | Electrical Output Piezoelectric Lane (W/Day) |
|---|---|---|---|---|
| 121 | 100 | 1 | 17.41 | 3.6 |
| 494 | 73.68 | 7 | 16.93 | 14.69 |
| 989 | 72.59 | 14 | 16.37 | 29.40 |
| 1487 | 71.46 | 21 | 15.8 | 44.21 |
| 1988 | 69.99 | 28 | 15.24 | 59.11 |
| 2495 | 67.19 | 37 | 14.52 | 74.18 |
| 2930 | 58.03 | 50 | 13.57 | 87.11 |

### 5.5. Proposal for the Energy Storage and ITS Technologies

In summary, the electrical output for the 10 m road are listed in Table 7.

**Table 7.** Electric power output.

| Solar Lane | Piezoelectric Lane | Tegs Lane |
|---|---|---|
| 16.1 kWh/day | 3.6 W/day | 2.18 Wh/day |

The results clearly show that the solar road is by far the best solution, in terms of the electric power output. Accordingly, all of the ITS could be easily charged using this technology. This brings up the question: why use piezoelectricity and TEGs? Beyond the cost-benefit considerations, piezoelectricity and TEGs could represent a valid alternative when the solar road is not implementable: for instance in tunnels, shaded roads, or areas that are poorly exposed to solar radiation.

As mentioned above, the solar road, TEGs, and piezoelectric materials provide intermittent electric energy. Therefore, the energy harvesting roads should be coupled with energy storage systems, installed along the roadside. According to the results of the case study, we propose the following batteries (Table 8):

**Table 8.** Type of batteries.

| Energy Source | Battery Type | Capacity |
|---|---|---|
| Solar lane | lead-acid or lithium | 6 kWh |
| Piezoelectric lane | nickel metal hydride (Ni-MH) | 2 Ah |
| TEGs lane | nickel metal hydride (Ni-MH) | 1 Ah |

With this case study, we are able to successfully calculate the estimate of the energy generated through each source. As indicated in Table 1, we have an estimate of how much energy is required for each ITS and it could help us to indicate the best energy harvesting techniques for each ITS application, and which could be the best practice for it. Table 9 shows some of the harvesting technologies that could be best suitable for various ITS applications. For the next generation of ITS, having efficient energy resources is essential for their business cases and implementation. Hence the coexistence of energy harvesting techniques and ITS is a pivotal step in the accomplishment of future roads and the expansion of their used cases.

**Table 9.** Best harvesting techniques suitable for various ITS technologies.

| ITS Application | Solar Roads | Piezoelectric Road | TEGs |
|---|---|---|---|
| CCTV surveillance | X | | |
| Sensor technology | X | X | |
| WIM technology | X | X | |
| Wireless sensor Technology | X | X | X |
| RFID Systems | X | X | |
| RSUs/Data acquisition systems | X | | |
| DMS/Portable chargeable message signs | X | | |

## 6. Advantages and Disadvantages of the Coexisting Systems

The utilisation of energy harvesting techniques are the way forward when it comes to using efficient and eco-friendly techniques of energy generation. Providing power to these ITS technologies is one of the main issues when it comes to installation. Hence novel energy harvesting techniques could be an efficient way for use in ITS.

However, the cost and power generation vary from one technique to other. It is therefore important to assess the merits and demerits of the individual techniques and their impact as a coexisting systems with ITS. Table 10 shows some of the advantages and disadvantages of these techniques, in conjunction with ITS technology.

Moving from the comparison of Table 10, the following observations can be drawn: (i) thanks to the high power output, the solar pavement can answer the power needs of any ITS; (ii) the piezoelectric road fits well with the ITS which must activate only in correspondence with the passage of a vehicle; (iii) due to the poor power output, TEGs can coexist only with wireless low-power sensors; (iv) piezoelectric and TEGs roads could be preferable to the solar roads, only in tunnels and shaded areas; (v) a common disadvantage of the coexistence between ITS and energy harvesting roads is the continuous energy supply. Therefore, the use of storage systems seems necessary, especially for the continuous monitoring.

**Table 10.** Advantages vs disadvantages of the coexistence between energy harvesting roads and ITS technologies.

| Technology & Energy Harvesting Road | Advantages | Disadvantages |
|---|---|---|
| All ITS technology & solar road | If the road is well exposed to solar radiation, the coexistence is always feasible | Must be combined with a storage system for the energy supply during night time |
| Sensor technology & piezoelectric road | Optimized use (the sensor detection activates in correspondence with the passage of the vehicle) | Because of the discontinuity of the energy supply, it is not adapted for continuous monitoring |
| WIM technology & piezoelectric road | Piezoelectric road can generate power at a low speed and match with the WIM technology | It should be combined with a storage system |
| RFID systems & piezoelectric roads | Optimized use (the RFID activates in correspondence with the passage of the vehicle) | RFID could be combined with a road side unit, which requires more energy |
| Wireless sensor technology & piezoelectric road | Piezoelectric road could fully meet the power needs of the wireless sensor | If the wireless sensor is used for continuous sensing, it must be combined with a storage systemWireless sensors could be combined with a road side unit, which requires more energy |
| Wireless sensor technology & TEGs | TEGs provide a continuous current, which fits well with the continuous monitoring of the wireless sensor | Wireless sensors could be combined with a road side unit, which requires more energy |

## 7. Conclusions

ITS are an emerging technology and their utilization has a tremendous impact on the transport infrastructures. However, it is still a challenge to provide an adequate power supply for these technologies, due to the location restrictions. This paper reviews the various state-of-the-art ITS technologies and the innovative concept of energy harvesting techniques, which pave the path for easy to install and efficient power sources. This paper also discusses that ITS and energy harvesting roads can coexist in harmony under certain constraints and limitations. Depending on the type of ITS, the most suitable energy harvesting technology can be selected by taking into account: (i) the energy consumption of the ITS; (ii) the operating conditions of the ITS (continuous or intermittent monitoring); (iii) the type of road; (iv) the traffic conditions; (v) the exposure to solar radiation.

The energy consumption can vary from 0.7 mW for the wireless sensors, up to 300 W/m$^2$ for the DMS/portable chargeable message signs. As emerged from the case study, the solar road is able to cover the power needs of all of the ITS. Furthermore, traffic heavily influences the power output of the energy harvesting pavements. In more detail, when the traffic flow increases from 121 to 2930 veh/hour, the power output of the solar lane decreases by up to 22%, while the power output of the TEGs and of the piezoelectric lane increases by 1.67 and 24 times, respectively. The huge sensitivity of the piezoelectric lane to the traffic is not surprising, because the power generation occurs at each passage of the vehicles' axles.

The main constraint is given by the intermittence of solar radiation. Therefore, the solar road must be equipped with lead-acid or lithium storage systems. TEGs and piezoelectric roads could be applied in tunnels or in shaded areas and they could be equipped with low capacity batteries (i.e., nickel metal hybrid). For low power ITS (sensors, WIM technology,

and RFID systems), piezoelectric roads could be the best option in heavily trafficked sections, especially if the ITS must activate only in correspondence to the passage of a vehicle. A problem may arise during the installation phase, which requires the temporary closure of the road. TEGs could be a valid alternative for wireless sensors in poorly trafficked sections, or to avoid the perturbation/disruption of the traffic, since they are installed along the roadside. In the presence of road side units, piezoelectric roads and TEGs would not be able to provide enough power, unless the road section is densely equipped over a large surface. In this scenario, the benefits deriving from the energy harvesting technologies would not justify the costs. Further research could move forward a detailed life-cycle cost analysis of certain ITS and energy harvesting roads (i.e., CCTV surveillance and solar roads), in the perspective of an "in-situ" application.

**Author Contributions:** Conceptualization, D.V., N.B. and G.F.; methodology, D.V., N.B. and G.F.; software, D.V.; validation, D.V., N.B. and G.F.; formal analysis, D.V., N.B. and G.F.; investigation, D.V., N.B. and G.F.; resources, D.V., N.B. and G.F.; data curation, D.V., N.B. and G.F.; writing—original draft preparation, D.V., N.B. and G.F.; writing—review and editing, D.V., N.B. and G.F.; visualization, D.V., N.B. and G.F.; supervision, D.V., N.B. and G.F.; project administration, D.V., N.B. and G.F.; funding acquisition, D.V. All authors have read and agreed to the published version of the manuscript.

**Funding:** This research received no external funding.

**Conflicts of Interest:** The authors declare no conflict of interest.

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
