# Peer review of "Coexistence of Energy Harvesting Roads and Intelligent Transportation Systems (ITS)"

_infrastructures, doi:10.3390/infrastructures8010014_

Round 1

Reviewer 1 Report

The paper “Coexistence of energy harvesting roads and intelligent transportation systems (ITS)” has some merits but needs the following revisions.

Authors may wish to focus on devices and methods used to assess structural health conditions (cf. 10.3390/electronics8101180).

In “Table 1. Energy consumption of ITS technologies” the differences between sensor technology and wireless sensor technology are quite difficult to understand. Please briefly discuss in the manuscript.

Table 2. It is not clear if thermoelectric generators are a part of solar cell technology or a different technology that can be coupled with solar roads. Please improve the description.

Section 5. If the traffic increases solar roads could produce less (?), TEG technology probably less (?), and piezo probably more (?). Did you consider the sensitivity of your results to traffic?

Author Response

Dear reviewer, thanks for your time and your interesting comments and observations. Please, find attached our answers.

Kind Regards

Reviewer 2 Report

It is of great significance and interest to examine the co-existence of ITS and energy harvesting pavement technology, which can figure out that whether the ITS can feed it by itself from the perspective of energy. Some specific comments were given below:

1) Page 3, Line 100: Please revise “advance” to “advanced”.

2) Subfigure number or code are missing in Figure 2.

3) Section 4: A schematic diagram is recommended to better present the working principle of energy storage system.

4) Why not take the TEDGs scenario into consideration in the case study section?

5) Section 5: Please specify the materials used for solar road and the type of energy harvesting elements used for the other two pavements.

6) Any consideration on the impacts of embedded energy harvesters on the service life (e.g., bearing performance) of pavements and the performance deterioration issue of these electronic elements over time? Especially for the solar pavement with the best energy generation performance.

Author Response

(The authors gave the same response as above.)
